# Simple is Better: Training an End-to-end Contract Bridge Bidding Agent without Human Knowledge

## Abstract

Contract bridge is a multi-player imperfect-information game where one partnership collaborate with each other to compete against the other partnership. The game consists of two phases: bidding and playing. While playing is relatively easy for modern software, bidding is challenging and requires agents to learn a communication protocol to reach the optimal contract jointly, with their own private information. The agents need to exchange information to their partners, and interfere opponents, through a sequence of actions. In this work, we train a strong agent to bid competitive bridge purely through selfplay, outperforming WBridge5, a championship-winning software. Furthermore, we show that explicitly modeling belief is not necessary in boosting the performance. To our knowledge, this is the first competitive bridge agent that is trained with no domain knowledge. It outperforms previous state-of-the-art that use human replays with 70x fewer number of parameters.

## 1 Introduction

Games have long been recognized as a testbed for reinforcement learning. Recent technology advancements have outperformed top level experts in perfect information games like Chess (Campbell et al., 2002) and Go (Silver et al., 2016; 2017), through human supervision and selfplay. During recent years researchers have also steered towards imperfection information games, such as Poker (Brown & Sandholm, 2018; Moravčík et al., 2017), Dota 2 [1], and real-time strategy games (Arulkumaran et al., 2019; Tian et al., 2017). There are multiple programs which focus specifically in card games. Libratus (Brown & Sandholm, 2018) and DeepStack (Moravčík et al., 2017) outperforms human experts in two-player Texas Holdem. Bayesian Action Decoder (Foerster et al., 2018b) is able to achieve near optimal performance in multi-player collaborative games like Hanabi.

Contract Bridge, or simply Bridge, is a trick-taking card game with 2 teams, each with 2 players. There are 52 cards (4 suits, each with 13 cards). Each player is dealt with 13 cards. The game has two phases: bidding and playing. In the bidding phase, each player can only see their own card and negotiate in turns via proposing *contract*, which sets an explicit goal to aim at during the playing stage. High contracts override low ones. Players with stronger cards aim at high contracts for high reward; while failing to reach the contract, the opponent team receives rewards. Therefore, players utilize the bidding phase to reason about their teammate and opponents' cards for a better final contract. In the playing phase, one player reveals their cards publicly. In each round, each player plays one card in turn and the player with best card wins the round. The score is simply how many rounds each team can win. We introduce the game in more detail in Appendix A.

Historically AI programs can handle the playing phase well. Back in 1999, the GIB program (Ginsberg, 1999) placed 12th among 34 human experts partnership, in a competition without the bidding phase. In more recent years, Jack [2] and Wbridge5 [3], champions of computer bridge tournament, has demonstrated strong performances against top level professional humans.

---

[1] `https://openai.com/blog/openai-five/`
[2] `http://www.jackbridge.com/eindex.htm`
[3] `http://www.wbridge5.com/`

On the other hand, the bidding phase is very challenging for computer programs. During the bidding phase a player can only access his own 13 cards (private information) and the bidding history (public information). They need to exchange information with their partners and try to interfere opponents from doing so through a sequences of non-decreasing bids. Moreover these bids also carry the meaning of suggesting a contract. If the bid surpasses the highest contract they can make, they will get negative score and risk of being doubled. Thus, the amount of information exchange is constrained and dependent on the actual hands. Nevertheless the state space is very large. A player can hold $6.35 \times 10^{11}$ unique hands and there are $10^{47}$ possible bidding sequences. Human has designed a lot of hand-crafted rules and heuristics to cover these cases, called bidding system, and designated a meaning to many common bidding sequences. However, due to large state space, the meaning of these sequences are sometimes ambiguous or conflicting. The bidding system itself also has room for improvement. The award winning programs often implement a subset of some specified human bidding system. Recently, there are also attempts to learn such a bidding system automatically through reinforcement learning. These methods either focus on bidding in the collaborative only setting, where both opponents will bid PASS throughout (Tian et al., 2018; Yeh & Lin, 2016), or heavily used human expert data for extra supervision (Rong et al., 2019).

In this work, we propose a system that is the state-of-the-art in competitive bridge bidding. It allows end-to-end training without any human knowledge through selfplay. We propose a novel bidding history representation, and remove any explicit modeling of belief in other agent's state, which are shown to be critical in previous works (Rong et al., 2019; Tian et al., 2018). We show that selfplay schedule and details are critical in learning imperfect information games. We use a much smaller model (about 1/70 in total parameters compared with previous state-of-the-art (Rong et al., 2019)), and reach better performance than the baselines (Rong et al., 2019; Yeh & Lin, 2016). Furthermore, we outperform world computer bridge championship Wbridge5 by 0.41 IMPs per board over a tournament of 64 boards. Finally, we show an interpretation of the trained system, and will open source the code, model, and experimental data we use.

## 2 RELATED WORK

Imperfect information games, especially card games, have drawn multiple researchers' attention. Prior works on two-player Texas Holdem mainly focus on finding the Nash Equilibrium through variations of counterfactual regret minimization (Zinkevich et al., 2008). Libratus (Brown & Sandholm, 2018) utilizes nested safe subgame solving and handles off-tree actions by real time computing. It also has a built-in self improver to enhance the background blueprint strategy. DeepStack (Moravčík et al., 2017) proposed to use a value network to approximate the value function of the state. They both outperform top human experts in the field. Bayesian Action Decoder (BAD)(Foerster et al., 2018b) proposes to model public belief and private belief separately, and sample policy based on an evolving deterministic communication protocol. This protocol is then improved through Bayesian updates. BAD is able to reach near optimal results in two-player Hanabi, outperforming previous methods by a significant margin.

In recent years there are also multiple works specifically focusing on contract bridge. Yeh and Lin (Yeh & Lin, 2016) uses deep reinforcement learning to train a bidding model in the collaborative setting. It proposes Penetrative Bellman's Equation (PBE) to make the Q-function updates more efficient. The limitation is that PBE can only handle fixed number of bids, which are not realistic in a normal bridge game setting. We refer to this approach as `baseline16`. Tian et al (Tian et al., 2018) proposes Policy Belief Learning (PBL) to alternate training between policy learning and belief learning over the whole selfplay process. PBL also only works on the collaborative setting. Rong et al (Rong et al., 2019) proposes two networks, Estimation Neural Network (ENN) and Policy Neural Network (PNN) to train a competitive bridge model. ENN is first trained supervisedly from human expert data, and PNN is then learned based on ENN. After learning PNN and ENN from human expert data, the two network are further trained jointly through reinforcement learning and selfplay. PBE claims to be better than Wbridge5 in the collaborative setting, while PNN and ENN outperforms Wbridge5 in the competitive setting. We refer to this approach as `baseline19`.

Selfplay methods have been proposed for a long time. Back in 1951, Brown et al (Brown, 1951) proposes fictitious play in imperfect information games to find the Nash Equilibrium. This is a classic selfplay algorithm in game theory and inspires many extensions and applications (Brown

& Sandholm, 2018; Heinrich et al., 2015; Heinrich & Silver, 2016; Moravčík et al., 2017). Large scale selfplay algorithms do not emerge until recent years, partially due to computation constraint. AlphaGo (Silver et al., 2016) uses selfplay to train a value network to defeat the human Go champion Lee Sedol 4:1. AlphaGoZero (Silver et al., 2017) and AlphaZero (Silver et al., 2018) completely discard human knowledge and train superhuman models from scratch. In Dota 2 and StarCraft, selfplay is also used extensively to train models to outperform professional players.

Belief modeling is also very critical in previous works about imperfect information games. Besides the previous mentioned card game agents (Foerster et al., 2018b; Rong et al., 2019; Tian et al., 2018), LOLA agents (Foerster et al., 2018a) are trained with anticipated learning of other agents. StarCraft Defogger (Synnaeve et al., 2018) also tries to reason about states of unknown territory in real time strategy games.

# 3 METHOD

## 3.1 PROBLEM SETUP

We focus on the bidding part of the bridge game. Double Dummy Solver (DDS) [4] computes the maximum tricks each side can get during the playing phase if all the plays are optimal. Previous works show that DDS is a good approximate to human expert real plays (Rong et al., 2019), so we directly use the results of DDS at the end of bidding phase to assign reward to each side. The training dataset contains randomly generated 2.5 million hands along with their precomputed DDS results. The evaluation dataset contains 100k such hands. We will open source this data for the community and future work.

Inspired by the format of duplicate bridge tournament, during training and evaluation, each hand is played twice, where a specific partnership sits North-South in one game, and East-West in another. The difference in the results of the two tables is the final reward. In this way, the impact of randomness in the hands is reduced to minimum and model's true strength can be better evaluated. The difference in scores is then converted to IMPs scale, and then normalized to [-1, 1].

## 3.2 INPUT REPRESENTATION

We encode the state of a bridge game to a 267 bit vector. The first 52 bits indicate that if the current player holds a specific card. The next 175 bits encodes the bidding history, which consists of 5 segments of 35 bits each. These 35 bit segments correspond to 35 contract bids. The first segment indicates if the current player has made a corresponding bid in the bidding history. Similarly, the next 3 segments encodes the contract bid history of the current player's partner, left opponent and right opponent. The last segment indicates that if a corresponding contract bid has been doubled or redoubled. Since the bidding sequence can only be non-decreasing, the order of these bids are implicitly conveyed. The next 2 bits encode the current vulnerability of the game, corresponding to the vulnerability of North-South and East-West respectively. Finally, the last 38 bits indicates whether an action is legal, given the current bidding history.

We emphasize that this encoding is quite general and there is not much domain-specific information. `baseline19` presents a novel bidding history representation using positions in the maximal possible bidding sequence, which is highly specific to the contract bridge game.

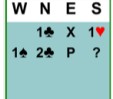

| Own Cards | Bid history, multi-hot of contract bids | | | | | Vulner-ability | Available Action |
|---|---|---|---|---|---|---|---|
| | Own | Partner | Left Opp | Right Opp | Double Indicator | | |
| 52 | 35 | 35 | 35 | 35 | 35 | 2 | 38 |

Figure 1: Input representation. With the decision point shown in the example, South will mark the following bits in the bidding history encoding: 1♡ in "Own" segment, 1♣ and 2♣ in "Partner" segment, 1♠ in "Left Opp" segment, and 1♣ in "Double Indicator" segment.

---

[4] https://github.com/dds-bridge/dds

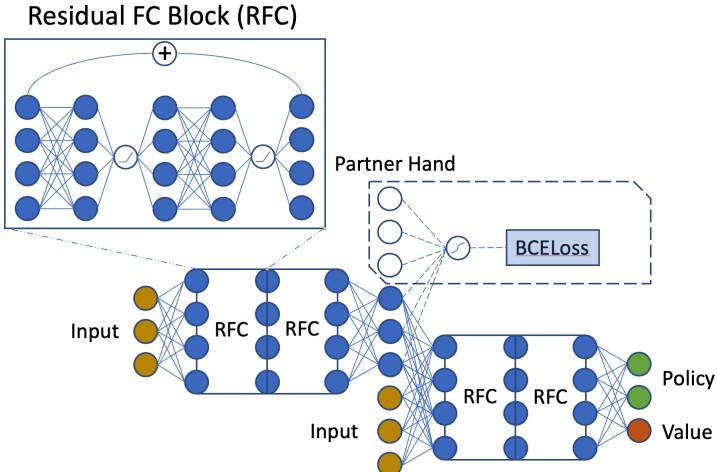

Figure 2: Network Architecture. Supervision from partner's hand is unused in the main results, and used in the ablation studies. BCELoss stands for Binary Cross Entropy Loss.

## 3.3 NETWORK

We use a similar network structure that is used in `baseline19`. As show in Figure 2, the network consists of an initial fully connected layer, then 4 fully connected layer with skip connections added every 2 layers to get a latent representation. We use 200 neurons at each hidden layer, so it is much smaller (about 1/70 in parameter size compared with `baseline19`). The full network architecture is shown In Figure 2. However, during our training we do not use partner's information to further supervise the belief training. We investigate the impact of training belief separately, and find that our model cannot benefit from extra supervision from partner's information.

From the latent representation, one branch is to a policy head. It is a fully connected layer to 38 output neurons, masking out illegal actions provided in the input, and then normalizes to a log policy. The other branch is a value head, which is just a fully connected layer to 1 neuron.

## 3.4 TRAINING DETAILS

**RL Method and Platform Implementation**. We use selfplay on random data to train our models. The model is trained with A3C (Mnih et al., 2016) using improved ELF framework (Tian et al., 2017). ELF supports off-policy training with importance factor correction, and has inherent parallelization implementations to make training fast. We implement contract bridge game logic and feature extraction logic in C++. Each game thread has 2 agent groups, namely training agent and opponent agent. Each agent group has a corresponding batcher. Once an agent needs an action, the current state and feature is sent to the batcher in ELF. ELF uses pybind to communicate between C++ and python. If batcher gathers enough data from different game threads for a specific actor group, the batch is forwarded to python for a pytorch model to evaluate. The results are then returned to the C++ game thread through pybind. ELF supports zero-copy during this process. During the selfplay training, the model of training agent actor group keeps updating, while the model of opponent agent actor group updates only when `opponent update frequency` condition is met. We implement an additional interface to track the full game trajectory. Once the game finishes, the interface receives a terminal flag with a reward signal. It fills all the history state / action pairs with the correct discounted rewards. This whole episode is then sent to the ELF train batcher to perform the actor critic update. The whole training process takes roughly 4-5 hours to converge on a single GPU.

**Training Parameters**. During training we run 500 games in parallel and use batch size of 100. We use an entropy ratio of 0.01 in A3C training. The discount factor is set to 1 to encourage longer sequences of information exchange, and since the bidding is non-decreasing, it will not cause convergence issue. We train the model using RMSProp with a learning rate of 1e-3. We fine tune our model by dropping learning rate to 1e-4 at 50k training minibatches and further dropping it to 1e-5 at 70k minibatches. During training we use multinominal exploration to get the action from a policy distribution, and during evaluation we pick the greedy action from the model.

Table 1: Performance Comparison. The left table compares performance when giving different weights to the belief loss and the performance when using the same history encoding as (Rong et al., 2019). The right table shows performance under different level of diversity of opponent models, by updating the opponent model at different frequency or sample opponent model randomly or using Nash Equilibrium.

A

| Ratio r | imps $\pm$ std |
|---|---|
| 0 | **2.31 $\pm$ 0.15** |
| 0.01 | 1.90 $\pm$ 0.29 |
| 0.1 | 1.63 $\pm$ 0.27 |
| 1 | 1.22 $\pm$ 0.22 |

| Hist encoding | |
|---|---|
| `baseline19` hist | 1.27 $\pm$ 0.22 |

B

| Update frequency | imps $\pm$ std |
|---|---|
| 1 | 2.26 $\pm$ 0.10 |
| 50 | 2.14 $\pm$ 0.20 |
| 100 | 2.08 $\pm$ 0.07 |
| 200 | **2.31 $\pm$ 0.15** |

| Opponent Diversity | |
|---|---|
| Randomly sample | 2.09 $\pm$ 0.04 |
| Nash averaging | 2.18 $\pm$ 0.20 |

Figure 3: Training curves for different update frequency. From left to right, the opponent model is updated every 1, 50, 100, 200 minibatches. Epoch is defined as 200 minibatches.

**Baselines**. As suggested by the authors of `baseline16`, we modify their pretrained model to bid competitively, by bidding PASS if the cost of all bids are greater than 0.2. We implement this and further fix its weakness that the model sometimes behaves randomly in a competitive setting if the scenario can never occur in a collaborative setting. We benchmark against them at each episode. We could not fully reproduce the results in `baseline19` so we cannot directly compare against them.

## 4 EXPERIMENTS

### 4.1 MAIN RESULTS

We train a competitive bridge bidding model through selfplay. We perform a grid search on hyper-parameters such as discount factor, exploring rate, learning schedules and find the best combination. The training curve against `baseline16` is shown in Figure 3. As can be seen, we significantly beat `baseline16` 2.31 IMPs per board. We manually run a 64 board tournament against Wbridge5, and outperforms it by 0.41 IMPs per board. The standard error over these 64 boards are 0.27 IMPs per board, which translate to 93.6% win probability in a standard match. This also surpasses the previous state-of-the-art `baseline19`, which outperforms Wbridge5 by 0.25 IMPs per board. It is shown in previous work that a margin of 0.1 IMPs per board is significant (Rong et al., 2019).

We outperform `baseline16` with a large margin partially due to `baseline16` cannot adapt well to competitive bidding setting. It can also only handle a fixed length of bids. We outperform `baseline19` mainly due to a better history encoding and not to model belief explicitly. These results are shown in the ablation studies.

### 4.2 ABLATION STUDIES

Prior works focus on explicitly modeling belief, either by adding an auxiliary loss to train jointly (Rong et al., 2019), or alternating stages between training policy and belief (Tian et al., 2018). However, training belief using supervision from partner's hand does not help in our model. We set the final loss as $L = rL_{belief} + L_{A3C}$. where $r$ is a hyper-parameter to control the weight on the auxiliary

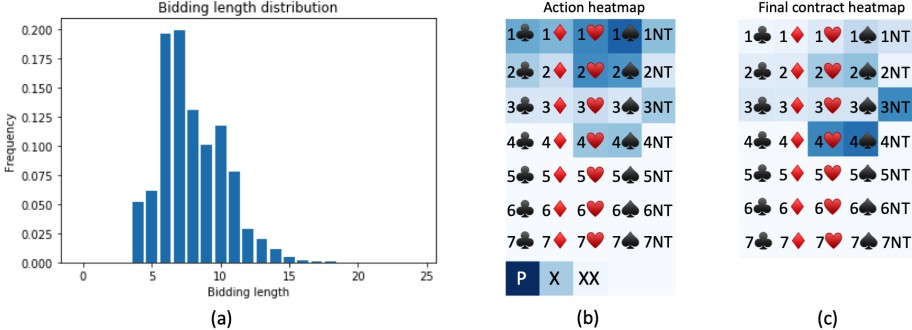

Figure 4: Statistical visualization. (a) Bidding length histogram. (b) Heatmap for all actions during the bidding. (c) Heatmap for final contracts reached.

task, As shown in Table 1, when $r = 0$, the model reaches the best performance and the performance decreases as $r$ increase. This demonstrates that focusing on the main task can achieve better results.

Bidding history encoding plays a critical role in model architecture. `baseline19` proposed a novel representation of sequenced bidding history, which listed all possible actions in a sequence and then labeled what has been used. We compared our representation to theirs. As shown in Table 1 our encoding can reach a better performance. The potential reason why our encoding performs better is that the intrinsic order of bridge bidding is already kept by the action itself, so there is no need to specify the sequence, and our encoding captures the owner of each action.

In imperfect information games, one common strategy is to use a pool of opponents to add diversity to the experiences. We also investigate this strategy in bridge. To increase the diversity, we set two ways: First, we maintain a model-zoo with 20 most recent models and then randomly sample the opponent model from this zoo; Second, we save the 20 models with best performance and sample using the Nash Averaging strategy (Balduzzi et al., 2018). We find self-play with opponent using the most recent model works best in terms of performance comparing to baseline models as shown in Table 1. One possible explanation is that bridge is a game with both competition and collaborations. Mixed strategy can mislead both opponents and partners, so a Nash Averaging strategy will not work well enough. Hence, using the most recent model is more suitable for such training.

Besides the strategy to choose opponent model, we also study the impact of opponent model update frequency. As can be seen from Table 1, the final performances are similar. However, the training curve Figure 3 shows different patterns. Using the exact the same model for selfplay opponent during the training shows the most stable results, especially at the early stage of the training. It is possibly due to the fast model progression during the early stage of the training. If selfplay opponent does not update frequent enough it cannot learn new knowledge.

## 5 INTERPRETATION

### 5.1 VISUALIZATION

It is interesting to visualize what the model has learned, and understand some rational behind the learned conventions. In Figure 4, we show the bidding length distribution and frequency of each bid used, as well as the distribution of final contracts. The results are averaged from our best 3 models. We can see that typically agents exchanges 6-10 rounds of information to reach the final contract. The agent uses low level bids more frequently and puts an emphasis on $\heartsuit$ and $\spadesuit$ contracts. The final contract is mostly part scores and game contracts, particularly often 3NT, $4\heartsuit$, $4\spadesuit$, and we observe very few slam contracts. This is because part scores and game contracts are optimal based on DDS for 87% of hands[5]. The model does not optimize to bid slam contracts, because it needs to hold a firm belief after longer rounds of information exchange to bid a slam contract, and the risk of not making the contract is very high.

---

[5]https://lajollabridge.com/Articles/PartialGameSlamGrand.htm

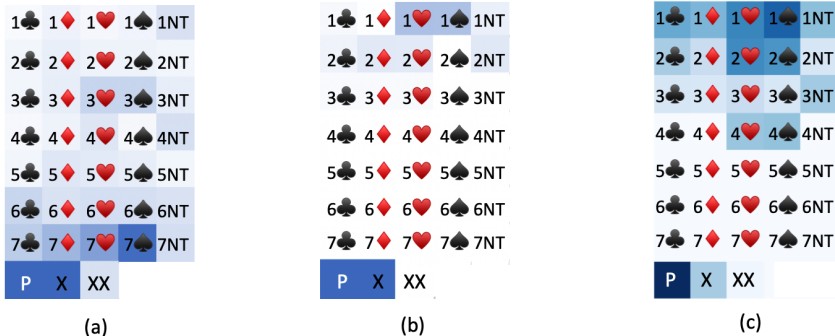

Figure 5: Action heatmaps for checkpoint models. (a) Early model. (b) Intermediate model. (c) Final model.

Table 2: Opening table comparisons. "bal" is abbreviation for a balanced distribution for each suit.

| opening bids | ours | SAYC |
|---|---|---|
| 1♣ | 8-20 HCP | 12+ HCP, 3+♣ |
| 1♦ | 8-18 HCP, 4+♦ | 12+ HCP, 3+♦ |
| 1♡ | 8-18 HCP, 4-6♡ | 12+ HCP, 5+♡ |
| 1♠ | 7-16 HCP, 4-6♠ | 12+ HCP, 5+♠ |
| 1NT | 14-18 HCP, bal | 15-17 HCP, bal |
| 2♣ | 8-13 HCP, 5+♣ | 22+ HCP |
| 2♦ | 7-11 HCP, 5+♦ | 5-11 HCP, 6+♦ |
| 2♡ | 7-11 HCP, 5+♡ | 5-11 HCP, 6+♡ |
| 2♠ | 7-11 HCP, 5+♠ | 5-11 HCP, 6+♠ |
| 2NT | 14+ HCP, 4+♣, 4+♦ | 20-21 HCP, bal |

## 5.2 BIDDING PATTERN EVOLUTION

It is important to be understand how the model evolves during the selfplay training. We pick three different checkpoint models along a single training trajectory, and check the frequency of each bid used. The result is shown in Figure 5. (a) is an early model. Since it behaves mostly randomly, and bids are non-decreasing, most contracts end at 6 or 7-level. This is clearly a very naive model. (b) is an intermediate model after about 10k minibatches training. The model learns that high level contracts are unlikely to make, and gradually starts to explore low level contracts that can make with the right hands. (c) is the final model which learns to prioritize NT and major contracts through information exchange and optimized categorization of various hands.

## 5.3 OPENING TABLE

There are two mainstream bidding system human experts use. One is called natural, where opening and subsequent bids usually shows length in the nominated suit, e.g. the opening bid 1♡ usually shows 5 or more ♡ with a decent strength. The other is called precision, which heavily relies on relays of bids to partition the state space, either in suit lengths or hand strengths. e.g. an opening bid of 1♣ usually shows 16 or more High Card Points (HCP)[6], and a subsequent 1♡ can show 5 or more ♠. To further understand the bidding system the model learns, it is interesting to establish an opening table of the model, defined by the meaning of each opening bid. We select one of the best models, and check the length of each suit and HCP associated with each opening bid. From the opening table, it appears that the model learns a semi-natural bidding system with very aggressive openings.

## 5.4 BIDDING EXAMPLES

We check a few interesting hands from the tournament between our model and Wbridge5. We present the following 5 examples in Figure 6.

---

[6]High Card Points is a heuristic to evaluate hand strength, which counts A=4, K=3, Q=2, J=1

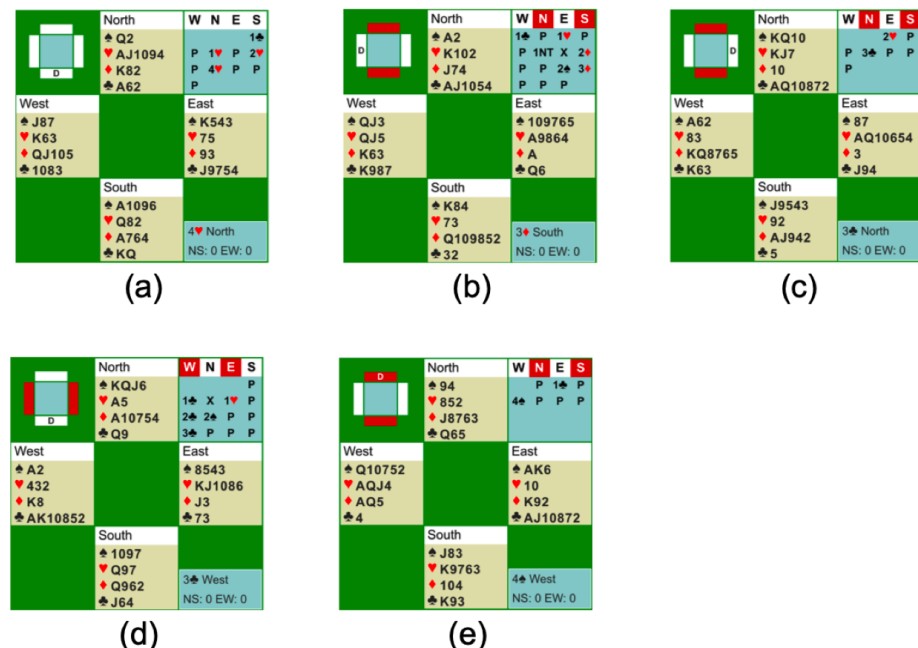

Figure 6: Bidding examples. D marks the dealer, and red seats indicate vulnerable side.

(a) This is a purely collaborative auction where our agents sit North-South. South chooses to open an artificial short ♣ suit. North shows his ♡ suit, and South raises his partner in return. With a strong hold North re-raise to 4♡, a game contract. The best contract determined by double dummy solver is 6♡, but it is due to the favorable position of missing honor cards, so it is not worth the risk to bid it.

(b) This is a competitive auction where our agents sit East-West. The first few bids are natural. East, holding 5 cards in both ♡ and ♠, takes action to double opponent's 1NT bid. While partner is silent East balances with 2♠ bid again. This successfully pushes opponents into a non-making 3♢ contract.

(c) Preemptive bids play an important role in bridge bidding. Historically a 2 level opening indicates a very strong hand, but modern bidding system bids it with a relative weak hand with a long suit (called weak 2). It is due to this hand type is much more frequent, and it can disrupt opponent's bidding by taking away the bidding space. In a standard system weak 2 usually promises 6 cards in the nominated suit, but from the opening table we can see that our agents do it more aggressively with routinely 5 cards. In this hand our agent opens a weak 2♡, and as a result North-South fails to find their best contract in ♠, and bids a non-making 3♣ instead, due to lack of information exchange.

(d) Double is a critical bid, unique to competitive bidding. The original meaning is that the doubler is confident to beat opponents' contract. However it can also be used to transmit other information. In this hand, our agent sitting North doubles opponent's 1♣ with a short ♣ suit and support for other suits, and a strong hand. This aligns well with the meaning of a modern "takeout double".

(e) The model jumps to 4♠ too quickly. While the contract is making, it fails to investigate slam opportunities on this hand.

## 6    CONCLUSION AND FUTURE WORK

In conclusion, we provide a strong baseline that is the state-of-the-art in bridge bidding, with a significantly smaller model. We offer insights through ablation studies to rethink about the training process and belief modeling in imperfect information games. We also interpret the learned bidding system through statistical visualizations and bidding examples. Our code, model and experimental data will be publicly available. We believe this addition is beneficial to the bridge community and imperfect information game researchers, to push forward further research in this direction. It remains a challenging problem to correctly model belief, to reason counterfactually, and to communicate efficiently in multi-agent imperfect information games. We leave this as future work.

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

## A  THE BRIDGE GAME

Bridge is played with a standard 52-card deck, and each player is dealt with 13 cards. There are two phases during the game, namely **bidding** and **playing**. After the game, **scoring** is done based on the bidding and playing. An example of contract bridge bidding and playing in shown in Figure 7.

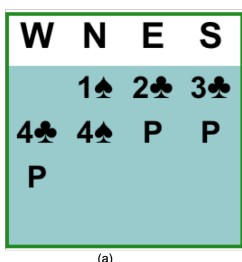
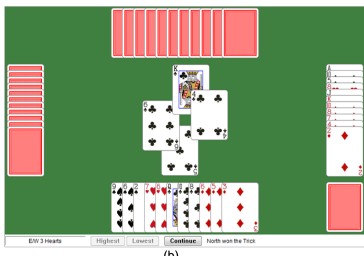

Figure 7: (a) A bidding example. North-South prevail and will declare the contract 4♠. During the bidding, assuming natural bidding system, the bid 1♠, 2♣, 4♣ and 4♠ are natural bids, which shows lengths in the nominated suit. The bid 3♣ is an artificial bid, which shows a good hand with ♠ support for partner, and shows nothing about the ♣ suit. To make the contract, North-South needs to take 10 tricks during the playing phase. (b) A playing example. Currently shown is the 2nd round of the playing phase. The dummy's card is visible to all players, and controlled by his partner, declarer. In the current round North player wins with ♣K, and will lead the next round.

**Bidding phase**. During the bidding phase, each player takes turns to bid from 38 available actions. The sequence of bids form an auction. There are 35 contract bids, which consists a level and a strain, ranging from an ordered set $\{1\clubsuit, 1\diamondsuit, 1\heartsuit, 1\spadesuit, 1NT, 2\clubsuit, ..7NT\}$ where NT stands for No-Trump. The level determines the number of tricks needed to make the contract, and the strain determines the trump suit if the player wins the contract. Each contract bid must be either higher in level or higher in strain than the previous contract bids. There are also 3 special bids. Pass (P) is always available when a player is not willing to make a contract bid. Three consecutive passes will terminate the auction (Four if the consecutive passes happen in the beginning of the auction), and the last contract bid becomes the final contract, with their side winning the contract. Double (X) can be used when either opponent has made a contract bid. It will increase both the contract score and the penalty score for not making the contract. Originally this is used when a player has high confidence that opponent's contract cannot be made, but it can also be used to communicate other information. Finally, Redouble (XX) can be used to further amplify the risk and reward of a contract, if the contract is doubled. Similarly, this bid can also be used to convey other information.

**Playing phase**. After the bidding phase is over, the contract is determined, and the owner of the final contract is the declarer. His partner becomes dummy. The other partnership is the defending side. During the playing phase, there are 13 rounds and each rounds the player plays a card. The first round starts with the defending side, and then dummy immediately lays down his cards, and the declarer can control both him and dummy. The trump suit is designated by the strain of the final contract (Or None if the strain is NT). Each round, every player has to follow suit. If a player is out of a certain suit, he can play a trump card to beat it. Discarding other suits is always losing in this round. The player with the best card wins a trick, and will play first in the next round. The required number of tricks to make the contract is contract level + 6. At the end of the game, if the declaring side wins enough tricks, they make the contract. Tricks in addition to the required tricks are called over-tricks. If they fail to make the contract, the tricks short are called under-tricks.

**Scoring**. if the contract is made, the declaring side will receive contract score as a reward, plus small bonuses for over-tricks. Otherwise they will receive negative score determined by under-tricks. Contracts below $4\heartsuit$ (except 3NT) are called **part score contracts**, with relatively low contract scores. Contracts $4\heartsuit$ and higher, along with 3NT, are called **game contracts** with a large bonus score. Finally, Contract with level 6 and 7 are called **small slams** and **grand slams** respectively, each with a huge bonus score if made. To introduce more variance, a vulnerability to randomly assigned to each board to increase both these bonuses and penalties for failed contracts. After raw scores are assigned the difference is usually converted to IMPs scale [7] in a tournament match setting, which is roughly proportional to the square root of the raw score, and ranges from 0 to 24.

---

[7]https://www.acbl.org/learn_page/howtoplaybridge/howtokeepscore/duplicate/

