# OpenReview forum: "Simple is Better: Training an End-to-end Contract Bridge Bidding Agent without Human Knowledge"
_ICLR.cc/2020/Conference — Reject_

### Official Review · AnonReviewer1 · 2019-10-23
**Official Blind Review #1**

**Rating:** 3

**Review:**

In recent years machine learning methods enjoy great popularity in the field of game theory. The paper presents a bidding agent for the card game contract bridge trained through selfplay. Contract bridge is an imperfect information game which has a bidding and a playing phase.
The bidding phase is challenging for computer agents because there are 1047 possible sequences and through the bidding process, the player has to exchange information with his game partner. The agent is trained using deep reinforcement learning without modeling of belief in other agent’s state or using human expert data.
The paper is easy to follow, it is almost free of errors and very well structured. The good readability of the text makes it easy to understand the key aspects of the work. The authors give a clear review of the related literature and work done on this field and use the approaches by Yeh & Lin (2016) and Rong et al. (2019) as baselines for their own method. The presented bidding agent is tested against Wbridge5 and outperforms the state-of-the-art approach by Rong et al. (2019). For a better comparison the authors also test the strategy of modelling belief but prove (controlling it via a hyper-parameter) that it doesn't help in training their model.
A strength of the paper is the interpretation of the training process and its statistical visualization. By investigating the bidding action in the training process the authors can show the learned strategy of the agent which is compared to the mainstream bidding systems of human experts.
The game of contract bridge is explained in the appendix which is a big plus so readers who didn't know this card game can follow the shown examples of bidding actions.
The authors plan to publish their code, the trained model and the experimental data so the results will be reproduceable for future work.

Overall, I have troubles in assessing the novelty of the system, since it is not my area of expertise. A concern is that there might be better fitting venues for this topic than ICLR.

**Experience Assessment:**

I do not know much about this area.

**Review Assessment: Checking Correctness Of Derivations And Theory:**

I did not assess the derivations or theory.

**Review Assessment: Checking Correctness Of Experiments:**

I assessed the sensibility of the experiments.

**Review Assessment: Thoroughness In Paper Reading:**

I read the paper at least twice and used my best judgement in assessing the paper.

---

### Official Review · AnonReviewer2 · 2019-10-24
**Official Blind Review #2**

**Rating:** 3

**Review:**

The authors propose a deep learning agent for automatic bidding in the bridge game. The agent is trained with a standard A3C reinforcement learning model with self-play, and the internal neural network only takes a rather succinct representation of the bidding history as the input. Experiment results demonstrate state-of-the-art performance with a simpler model. The authors discuss some findings with the proposed agent, such as the lack of need to explicitly model the belief and the possibility to self-train with different variants of opponents. Some visualization is also provided to understand how the trained agent behaves.

It is recommended to weak-reject the paper. There is an obvious contribution of using a simple model to reach state-of-the-art performance. The only concern is whether such a contribution is sufficient to warrant acceptance for a top conference. While setting a new milestone for the community is important (and having open-source code and data in the future will lead to a huge impact for the community), the current paper does not appear to have introduced sufficient "other contributions." In particular,

(1) The ablation study is very shallow. Section 4.2 shows four things: [a] training with an auxiliary belief task does not help the proposed agent; [b] simple representation is sufficiently good; [c] using a pool of opponents won't do better than using the most recent model as opponent; [d] frequency of updating the opponents does not matter much.

For [a], it is certainly interesting. But one could get to a deeper understanding by, for instance, zooming in to a ratio of 10^-3 or less, to see if there are sweet points of the choice of r. One could also get to a deeper understanding by checking how well BCELoss is doing, and how well the value loss is doing, to understand the trade-off. It is not even clear what the ranges of BCEloss and the value loss are, making it uncertain on whether r is properly chosen.

For [b], deeper study could be taken for analyzing "why", especially given that the more complicated encoding is significantly worse. The hand-waving explanation of "The potential reason why our encoding performs better is that the intrinsic order of bridge bidding is already kept by the action itself ..." at best suggests that the information between the baseline19 encoding and the author's are of the same information amount, but does not say why baseline19 is significantly worse. The authors leave a big question mark here that does not seem to match the grand title of "simple is better" of this paper.

For [c], deeper study could be taken for analyzing "what if some worse opponents are included" or "what if some targeted opponents are included." For instance, what if the authors include early-stage models to increase diversity? What is the trade-off between opponent capability/diversity and performance? What if using baseline16 as opponents---would the agent then beat baseline16 more easily? Note that baseline16 and the proposed agent basically learned a semi-natural bidding system. What if a precision system is set as a potential opponent---would the learned policy be very different?

For [d], getting a result that the update frequency leads to similar performance can hardly match the claim "We show that selfplay schedule and details are critical in learning imperfect information games." Also, the big error bar and the inconclusive trend in Table 1(B) does not answer the question on what frequency should best be used. Deeper discussions shall be included.

(2) Besides the ablation study, there is very little information about the design choices. For instance, why does the authors choose the specific RFC block? Would a full FC block do better or worse? Would a shallower network do better or worse? Actually, the title "simple is better" may be misleading, as the authors did not study even simpler choices. For RL algorithm, why A3C but not other algorithms?

(3) The authors' finding that the agent is conservative and does not bid high appears interesting. Nevertheless, it perhaps suggests that the authors have not compared with strong bidding systems (like precision) that aimed for bidding high to win more. That is, the claimed advantages are over semi-natural bidding systems. More study on whether the agent should bid high could greatly enrich this paper.

===

I've read the rebuttal. I thank the authors for the discussion on some possibilities in deepening the paper. I suggest the authors to avoid some hand-waving claims such as "We argue it is easier to learn the connections between them." In general, some of the claims (such as "simple is better" in the title) lacks sufficient scientific support, and I believe that this paper will have more impacts if it focuses more on supporting the claims with science evidence, not with human descriptions.

**Experience Assessment:**

I have published one or two papers in this area.

**Review Assessment: Checking Correctness Of Derivations And Theory:**

I assessed the sensibility of the derivations and theory.

**Review Assessment: Checking Correctness Of Experiments:**

I carefully checked the experiments.

**Review Assessment: Thoroughness In Paper Reading:**

I read the paper thoroughly.

---

> ### Author Response · Authors · 2019-11-12
> **Reply to R2**
>
> We thank the reviewer for the insightful feedbacks.
> We address reviewer's concerns in (1) as follows:
> (a)
> We run more experiments with smaller values of r. Picking r=1e-3 we can get 2.24±0.13.
> From previous logs, we can see that after the first 200 minibatches, while the policy is still quite random, the value loss is around 0.3~0.4 and the BCELoss (belief loss) is around 0.6. Since they are roughly on the same order of magnitude, a natural starting point is r=1. (Inspired by the original AlphaZero choice to directly add up policy loss and value loss). We observe that near convergence the value loss is about 0.02 and the belief loss is about 0.4. We argue that during the later part of training the belief loss become significant that the model shifts from its primary targets to the auxiliary task, even with smaller values of r, thus, it can improve to a certain stage and the performance will plateau.
> (b)
> The proposed bidding history representation is a more natural representation of history actions, and it has a direct one-to-one mapping to actions. We argue it is easier to learn the connections between them. In contrast, baseline19’s representation is quite sophisticated and uses human knowledge, and it can only be used in that specific game.
> (c)
> Training against baseline16 will not be Zero since it uses human knowledge. In practice, we observe that since baseline16 is a reasonably strong opponent, the agent cannot get rewards, and thus cannot improve.
> (d)
> Stated in end of section 4.2, we conclude that final performances are similar. We observe that using a higher frequency (f=1) will improve much faster in the earlier stage in the training, shown in Figure 3. This makes sense because the agent is able to iterate more to improve from the naive strategies. Thus we recommend to use f=1. We will make revisions to make it clearer.
>
> (2) Besides the ablation study, there is very little information about the design choices. For instance, why does the authors choose the specific RFC block? Would a full FC block do better or worse? Would a shallower network do better or worse? Actually, the title "simple is better" may be misleading, as the authors did not study even simpler choices. For RL algorithm, why A3C but not other algorithms?
> We did some ablation study on this. If we use a full FC block, the performance is actually much worse. It cannot even reach positive reward against baseline16. If we use a shallower network, e.g. use half of the RFC blocks, the performance is slightly worse at 2.29 ± 0.17. We will update these numbers in the next revision.
>
> (3) The authors' finding that the agent is conservative and does not bid high appears interesting. Nevertheless, it perhaps suggests that the authors have not compared with strong bidding systems (like precision) that aimed for bidding high to win more. That is, the claimed advantages are over semi-natural bidding systems. More study on whether the agent should bid high could greatly enrich this paper.
> It is not feasible to compare against precision systems, since there is no open source implementation of a precision system. We argue that our superior model is quite general in the following sense:
> 1 Systems are not just bipolar to be either natural or precision. E.g. in precision system there are quite a lot of natural bids, and in natural system, the asking bids and step responses are inspired from precision.
> 2 In fact, baseline16 is much closer to a precision system. From Table 7 in https://arxiv.org/pdf/1607.03290.pdf , we can see that 1c is a general strong bid and other bids are transfer openings, e.g. 1d shows hearts, 1h shows spades, 1s shows balanced opening.
> 3 Default Wbridge5 uses SAYC which is a mostly natural system.
>
> We agree that study on if agent can bid high is a great addition to this paper. However, bridge conventions are all about trade-offs. If you bid more precisely on hands where high level contracts can be achieved, the precision on lower level contracts(part scores and games) will decrease. For example, if you set 1c to be 20 or more points, then when it comes up it is much easier to bid high level contracts. But this will negatively impact the ability to reach the best low level contracts, especially in clubs. The agent should just maximize the expected payoff of the learned system.

---

### Official Review · AnonReviewer3 · 2019-10-24
**Official Blind Review #3**

**Rating:** 3

**Review:**

Simple is Better
--------------

This paper develops a method to train agents to bid competitively in the game of Bridge. The authors focus on the bidding phase of the game and develop a model to predict the best bid to make at each turn of the phase. The difficulty in the bidding lies in understanding the signals provided by your own teammate as well as the opponent team in order to estimate the state of the game, which is partially observable since each player cannot see the hands of the three other players. The authors show that explicitly modeling the belief of other agents is not necessary and that competitive performance can be achieved with self-play.

While the application is interesting, there is not much novelty in the method and factoring that, the empirical study is lacking as well.

Pros:
1. Interesting application of self-play to a multi-agent setting with limited communication!

Cons:
1. There doesn't seem to be much novelty method-wise other than the input representation of the game state. Given that there is an oracle which provides information on how the game would progress assuming optimal agents, could this even be turned into a standard classification problem?
2. It is not clear what the baselines being compared to are (baseline16 and baseline19). Also, the training curves are missing error bars. On the same lines, how many runs are the 'std' numbers in Table 1 reported over?


Other comments:
1. The authors emphasize that the method is not 'domain-specific'. Given that they build the representation using knowledge of bridge, I'm not sure if this is an appropriate statement to make.
2. The results section is a bit unclear. First, what are the numbers in table 1? The authors mention 'performance' but are these the number of tricks won or the difference between what the team bid and the actual number of tricks won? The baselines are unclear as well, so it is hard to evaluate the results.

**Experience Assessment:**

I have read many papers in this area.

**Review Assessment: Checking Correctness Of Derivations And Theory:**

N/A

**Review Assessment: Checking Correctness Of Experiments:**

I assessed the sensibility of the experiments.

**Review Assessment: Thoroughness In Paper Reading:**

I read the paper at least twice and used my best judgement in assessing the paper.

---

> ### Author Response · Authors · 2019-11-12
> **Reply to R3**
>
> We thank the reviewer for the insightful feedbacks.
>
> We argue there is more novelty than what R3 proposes.
> “Given that there is an oracle which provides information on how the game would progress assuming optimal agents, could this even be turned into a standard classification problem?”
> This is definitely not the case. The oracle, DDS, only provides optimal information during the playing phase, which we omit in the training of bidding phase. The oracle also assumes perfect information which is not the case during playing. DDS only sets up a meaningful reward for RL agent to optimize for the bidding phase.
> The bidding phase is much harder. First it is not possible to run supervised training on it - human and other bots’ data are based on different conventions or communication protocols. For example, the bid “1H” might mean “strong hearts” or “strong spades” depending on the protocol. Arguably, these human-designed heuristics are limited in their power and suboptimal.
> At a high level, each bidding sequence (order of 10^47) needs to be assigned a meaning and to form a meaningful partition of the whole state space (order of 10^12), with imperfect information. Furthermore agents need to cooperate and compete against each other. Using AlphaZero selfplay to address this problem is novel on its own, since AlphaZero previously are used with perfect information games only.
>
> “It is not clear what the baselines being compared to are (baseline16 and baseline19).”
> It is explained in Related Work (Section 2), and briefly explained again in Training Details (3.4)
> baseline16 uses Penetrative Q-Learning to form a collaborative baseline for bridge bidding. The authors also mention the method can be slightly modified to serve as a baseline for competitive bidding.
> baseline19 trains separate networks for policy and estimation/belief learning. They start with supervised training and then do selfplay with learned models.
>
> Other:
> "On the same lines, how many runs are the 'std' numbers in Table 1 reported over?"
> Std number are based on an average of 5 runs.
>
> "The authors emphasize that the method is not 'domain-specific'. Given that they build the representation using knowledge of bridge, I'm not sure if this is an appropriate statement to make."
> The domain-agonistic AlphaZero selfplay training also needs to capture the game state: for Go, the agent needs to know on a 19x19 board, the locations of the black and white stones; for Chess, where are all the pieces on a 8x8 board, and different pieces are encoded differently. The term “not domain-specific” means that it doesn’t encode domain knowledge (e.g., for Chess, Queen is more valuable than a Pawn, for Go, some stones are live/dead, territory is important) during the training. Likewise, in this paper, we claim that our method is not domain-specific”, because it does not feed domain specific knowledge such as what contracts are likely to make and with higher scores, and what each of the bids should mean. We let the selfplay agents to optimize and find out.
>
> "The results section is a bit unclear. First, what are the numbers in table 1? The authors mention 'performance' but are these the number of tricks won or the difference between what the team bid and the actual number of tricks won? The baselines are unclear as well, so it is hard to evaluate the results."
> The results are in IMPs scale, against baseline16, which is pointed out in 3.4 “baselines”, “We benchmark against them at each episode.” and 4.1 “main results”. The match is in duplicate setting, e.g. 2 identical deals are given, our agent sits in NS of Table 1 and EW of table 2. We calculate the score difference of two tables to minimize variance.
> We will make it more clear in the next revision.

---

### Author Response · Authors · 2019-11-15
**Rebuttal**

We thank the reviewer for the insightful feedbacks.

We have responded the detailed questions in individual comments.

We want to emphasize that this paper significantly advances the state-of-the-art of contract bridge bidding, which is a multi-agent imperfect information game. Furthermore, this is the first time that the algorithm learns from the full bidding phase and without human data. Prior works uses human data and feature/reward engineering extensively, and are restricted to 2-player games. The success to establish a benchmark in this field will attract researchers in this domain to move away from heuristic methods, and to focus more on generalized machine learning methods. Furthermore, it will also benefit the contract bridge community to explore a new bidding strategy learned by the agents.

---

### Decision · Program_Chairs · 2019-12-19

**Decision:**

Reject

**Comment:**

This paper proposes a new training method for an end-to-end contract bridge bidding agent. Reviewers R2 and R3 raised concerns regarding limited novelty and also experimental results not being convincing. R2's main objection is that the paper has "strong SOTA performance with a simple model, but empirical study are rather shallow."

Based on their recommendations, I recommend to reject this paper.